# The Role and Mechanism of Gut Microbiota in Pulmonary Arterial Hypertension

**DOI:** 10.3390/nu14204278

**Published:** 2022-10-13

**Authors:** Yi-Hang Chen, Wen Yuan, Liu-Kun Meng, Jiu-Chang Zhong, Xiao-Yan Liu

**Affiliations:** 1Heart Center and Beijing Key Laboratory of Hypertension, Beijing Chaoyang Hospital, Capital Medical University, Beijing 100020, China; 2Medical Research Center, Beijing Institute of Respiratory Medicine and Beijing Chaoyang Hospital, Capital Medical University, Beijing 100020, China; 3Department of Cardiology, Beijing Chaoyang Hospital, Capital Medical University, Beijing 100020, China; 4State Key Laboratory of Cardiovascular Disease, Fuwai Hospital, National Center for Cardiovascular Disease, Peking Union Medical College, Chinese Academy of Medical Sciences, Beijing 100032, China

**Keywords:** pulmonary arterial hypertension, gut microbiota dysbiosis, pulmonary vascular remodeling, metabolism, prebiotics and probiotics, microbiota transfer therapy

## Abstract

Pulmonary arterial hypertension (PAH) is a malignant pulmonary vascular disease characterized by increased pulmonary vascular resistance, pulmonary vasoconstriction, and right ventricular hypertrophy. Recent developments in genomics and metabolomics have gradually revealed the roles of the gut microbiota (GM) and its metabolites in cardiovascular diseases. Accumulating evidence reveals that the GM plays important roles in the occurrence and development of PAH. Gut microbiota dysbiosis directly increases the gut permeability, thereby facilitating pathological bacterial translocation and allowing translocation of bacterial products such as lipopolysaccharides from the gut into circulation. This process aggravates pulmonary perivascular inflammation and exacerbates PAH development through the endothelial–mesenchymal transition. Additionally, a shift in the composition of PAH also affects the gut metabolites. Changes in gut metabolites, such as decreased short-chain fatty acids, increased trimethylamine N-oxide, and elevated serotonin, contribute to pulmonary perivascular inflammation and pulmonary vascular remodeling by activating several signaling pathways. Studies of the intestinal microbiota in treating pulmonary hypertension have strengthened linkages between the GM and PAH. Probiotic therapy and fecal microbiota transplantation may supplement existing PAH treatments. In this article, we provide new insight for diagnosing, preventing and treating PAH by adding to the current knowledge of the intestinal flora mechanisms and its metabolites efficacy involved in PAH.

## 1. Introduction

Pulmonary arterial hypertension (PAH) is a malignant pulmonary vascular disease characterized by augmented pulmonary vascular resistance, pulmonary vasoconstriction, and right ventricular hypertrophy, ultimately resulting in right heart failure and death. The main pathophysiological mechanisms of PAH are endothelial injury, pulmonary vascular remodeling, and orthotopic thrombosis [1]. Owing to the lack of screening methods and biomarkers for early detection, patients with PAH often develop severe right ventricular dysfunction at the time of diagnosis [2]. The 1-, 3-, and 5-year survival rates for patients with PAH are 90.4%, 76.2%, and 65.4%, respectively, thereby enhancing the society economic burden in China and Western [3]. Moreover, current pharmacological therapies are expensive and non-curative, only modestly increasing patients’ exercise capacity and reducing hospital admissions [4]. Since 2005, no new therapeutic pathways for treating PAH have been identified [5]. Therefore, new approaches are urgently needed for early diagnosis and more efficient curative treatment for patients with PAH.

In the intestinal system, many bacteria, archaea, protists, fungi, and viruses form an ecological community known as the gut microbiota (GM). The microbiome produces plenty of metabolites with biological effects, thereby maintaining gut barrier function and homeostasis [6,7]. Gut microbes regulate digestion, metabolism, and innate immunity and prevent colonization of pathogens, while the gut supplies them with the proper environment for survival [8,9,10,11].

Dysbiosis of the GM has been described in multiple cardiovascular diseases, including atherosclerosis, hypertension, and platelet hyperactivity [12]. Accumulated evidence has indicated that gut integrity, the composition of GM, and its metabolites play important roles in the development of systemic inflammation, therefore modulating various cardiovascular disease risk factors. For example, the gut microbiota metabolite trimethylamine N-oxide (TMAO) directly constricts arterioles and sensitizes blood pressure levels in Ang II-induced hypertensive mice. [13]. At the same time, TMAO exerts pro-atherosclerotic effects via a variety of mechanisms [14]. In contrast, short-chain fatty acids (SCFAs) protect from cardiac damage and reduce atherosclerosis in experimental hypertension [15]. However, for PAH, a systemic disease involving many organs in addition to the lungs, such as the central nervous system, immune system, bone marrow, and intestines, the GM remains an understudied area [16]. Recent research indicates an association between the gut microflora and PAH, sparking great interest among scholars. On the one hand, persistent pulmonary hypertension leads to right ventricular hypertrophy, right heart failure, and systemic venous congestion, which alters the intestinal flora; on the other hand, changes in the GM greatly affect the intestinal barrier function [17]. When the gut barrier is impaired, intestinal Gram-negative bacteria release lipopolysaccharides (LPS) into the bloodstream, causing Toll-like receptor 4 (TLR4)-mediated endotoxemia [18,19], which induces pulmonary perivascular inflammatory responses and thus promotes the development of PAH. Furthermore, variations in gut microbial metabolites facilitate perivascular inflammation and the proliferation of pulmonary vascular smooth muscle cells, thereby initiating lung injury and pulmonary vascular remodeling, and finally leading to PAH [20,21,22] (Figure 1). However, the specific interaction between gut dysbiosis and PAH has not been clearly clarified. This review explores the relationship among intestinal flora, gut microbial metabolites, and PAH, and we discuss potential therapeutic strategies for PAH by reconstructing the GM.

## 2. The Gut Microbiota and Pulmonary Arterial Hypertension

### 2.1. Gut Microbiota Composition

The five predominant phyla in the human gut microbiome are *Bacteroidetes, Firmicutes, Actinobacteria, Proteobacteria,* and *Verrucomicrobia* [23]. In healthy guts, anaerobic *Bacteroidetes* and *Firmicutes* account for >90% of the total bacterial species, whereas *Proteobacteria* and *Actinobacteria* are the most abundant phyla in the circulating microbiome [24]. The number of genes in these microorganisms exceeds the total number of human genes by >100 times, greatly enriching the gene pool and metabolic functions of their human hosts. A stable intestinal microbiome is fundamental for preserving the intact epithelial barrier and preventing microbiota constituents from translocating to the epithelium, spreading to and infecting other distant areas of the body [25]. However, the GM differs between healthy and unhealthy individuals [16]. Gut microbiota activity and composition are leading factors influencing human health and causing diseases [26]. Diet, drugs, and lifestyles can affect and cause disproportions in the intestinal flora. Gut dysbiosis occurs when the number of beneficial bacteria decreases or the number of harmful bacteria increases, causing an imbalance between the GM and host, and eventually leading to diseases such as diabetes mellitus, malignant tumors, hypertension, and pulmonary hypertension [27,28].

### 2.2. Shifts in the Gut Microbiota Composition in Pulmonary Arterial Hypertension

Recent reports suggest a link between PAH and the GM. Gut microbiota dysbiosis was observed in rat models of experimental PAH, including Sugen 5416-induced, chronic hypoxia-induced, and monocrotaline (MCT)-induced rat models [29,30]. Similarly, intestinal microbiota imbalances have been detected in patients with pulmonary hypertension [31,32]. Long-term PAH can result in chronic right heart failure and systemic venous congestion. Congestion of the intestinal veins leads to reduced bowel perfusion, increased intestinal permeability, and gut bacterial and/or endotoxin translocation [33,34]. With the emerging role of the GM in health and diseases, the gut–lung axis provides new insight into the PAH pathogenesis and may be a novel therapeutic target for PAH.

Experimental animals induced via hypoxia/Sugen 5416 or intraperitoneal injection of MCT are widely used animal models of PAH. Wistar rats treated with hypoxia/Sugen 5416 exhibited an increased *Firmicutes*-to-*Bacteroidetes* ratio, a hallmark of gut dysbiosis, with a lower abundance of *Bacteroidetes* [29]. Moreover, gut genera that function as acetate and butyrate producers were depleted by hypoxia/Sugen 5416 and showed decreased serum acetate levels [29]. Monocrotaline-induced PAH rats also showed disordered gut microbiota [30]. It was found that compared with the control group, MCT-induced PAH rats showed higher relative abundances of *Firmicutes*, *Proteobacteria*, and *Actinobacteria* but lower relative abundances of *Bacteroidota* and *Spirochaetota*. Moreover, modifying the gut microbiota with antibiotics suppressed the increase in right ventricular systolic pressure, right ventricular hypertrophy, and pulmonary vascular remodeling in hypoxia/Sugen 5416-induced rats [28].

Consistent with the results from PAH animals, patients with PAH also exhibit altered gut microbiome profiles. Using metagenomic analyses, Kim et al. revealed the fecal microbial features of 18 patients with type I PAH and 13 controls without PAH [31]. Alpha diversity and bacterial richness and evenness were reduced in the intestinal flora of patients with type I PAH [31]. *Bifidobacterium*, a genus of acetate producers, was overrepresented, whereas *Coprococcus, Butyrivibrio, Lachnospiraceae*, and *Eubacterium*, which are butyrate-producing bacteria, and *Akkermansia* and *Bacteroides*, which are propionate-producing bacteria, were reduced in patients with type I PAH [31]. A subsequent study on 11 patients with chronic thromboembolic pulmonary hypertension (CTEPH) and 22 matched controls showed that patients with CTEPH had different gut microbiota compositions [32]. The butyrate-producing bacteria, *Roseburia* and *Faecalibacterium*, were also significantly reduced in patients with CTEPH [32]. Overall, these studies showed dysregulated gut microbiota in both animals and patients of PAH (Table 1), which may contribute to the development of PAH.

### 2.3. Lipopolysaccharides in Pulmonary Arterial Hypertension

The GM plays a major role in maintaining the normalization of intestinal permeability [7]. The intestinal mucosal layer, a component of the intestinal barrier, covers the intestinal cells like a shell and protects them from mechanical, chemical, and biological attacks [37]. The GM is essential for the formation of a proper mucosal layer, and its composition notably influences the intestinal mucus [38]. In a study with germ-free mice, researchers could experimentally aspirate only a few filled goblet cells and a small volume of intestinal mucus, because that mucus fails to mature and develop into the proper structure in the absence of bacteria or their metabolites [39]. Imbalanced intestinal flora in PAH rats alters the intestinal permeability, including fewer mucin-producing goblet cells, shortened villus lengths, and increased intestinal fibrosis and muscular tissue [40]. Decreases in SCFA-producing bacteria, such as *Coprococcus*, *Butyrivibrio*, *Lachnospiraceae*, *Eubacterium*, and *Clostridia*, in patients with PAH may epigenetically modify gut epithelial cells and consequently increase gut permeability. Patients with PAH exhibit increased plasma biomarkers for gut leakiness and inflammation, indicating that impaired mucosal barriers and increased permeability exist in these patients [6].

Increased gut permeability caused by gut dysbiosis allows commensal bacteria to translocate from the enteric cavity into circulation and promotes the generation of peripheral blood bacterial products, such as LPS. Lipopolysaccharides, also known as endotoxins, are the main structural components of Gram-negative bacteria and strongly affect foreign pathogen detection by host cells. Lipopolysaccharide levels were significantly increased in CTEPH-, type I PAH-, and MCT-induced animals [29,31,32]. As a pathogen-associated molecular pattern, LPS is recognized by pathogen recognition receptors, including TLRs. TLR4, the major signaling receptor for LPS, plays a vital role in LPS-induced inflammatory responses [41].

Toll-like receptor 4 is expressed in lung parenchymal and nonparenchymal cells, including endothelial cells (ECs), smooth muscle cells (SMCs), macrophages, and platelets. Toll-like receptor 4 and its downstream signaling pathways participate in PAH pathogenesis [42]. Several mechanisms underlie the effect of TLR4 activation on PAH development. As a component of innate immunity, TLR4-induced inflammation is an important mechanism triggering PAH. Lipopolysaccharides stimulate macrophages through TLR4 and initiate a series of signaling events, including activation of IκB kinase-activated nuclear factor κ-light-chain-enhancer of activated B cells (NF-κB), the mitogen-activated protein kinase (MAPK) family c-Jun N-terminal kinases, extracellular signal-regulated kinases (ERKs), and p38 [43]. Therefore, in chronic hypoxia-induced PAH rats, the depletion of inflammatory macrophages prevented remodeling of the pulmonary vascular extracellular matrix and development of pulmonary hypertension [44,45].

Activation of NF-κB, the central switchboard of inflammation, results in the release of cytokines (e.g., interleukin [IL]-4, IL-13, tumor necrosis factor-α, and IL-1β) and chemokines (e.g., IL-6 and IL-8) [46], which promote the development of PAH by regulating pulmonary vascular remodeling. In bone morphogenetic protein receptor type II-deficient mice chronically treated with LPS, TLR4 was upregulated in pulmonary arterial SMCs (PASMCs) and the lungs, and thus stimulated IL-6 and IL-8 production and led to pulmonary vascular remodeling [47]. Cytokines directly control pulmonary vascular cell proliferation, migration, and differentiation. For example, IL-6 overexpression promoted the growth and proliferation of porcine aortic ECs and PASMCs, and IL-6 knockout mice were resistant to hypoxia-induced PAH [48].

In addition to inflammation, TLRs on ECs also contribute to PAH development. ECs are widely involved in PAH-related pathological processes, including vasoconstriction, inflammation, cell viability, cell growth, and cell differentiation [49]. The effects of TLR signaling on endothelial dysfunction are multifaceted. TLR4 signaling increases the levels of vascular endothelial growth factors, which are central angiogenic molecules that regulate EC proliferation, migration, and tube formation during angiogenesis [50]. In addition, TLR signaling promotes PAH development through endothelial–mesenchymal transition (EMT), a process by which ECs acquire a mesenchymal cell phenotype. Increasing studies have established that EMT gives rise to the dysfunctional endothelial phenotype in patients with PAH [51,52,53]. Toll-like receptor 4 activation enhanced EMT and promoted the expression of the progenitor cell marker c-kit (CD117) in mouse pulmonary ECs, indicating a potential connection between TLR4 and EMT [54].

Furthermore, TLRs on platelets may also play important roles in PAH. After bonding to TLR4, LPS promoted P-selectin, IL-1β, and ATP expression in platelets, which may promote in situ thrombosis, a feature of human idiopathic PAH [55,56]. Genetic deletion of TLR4 on platelets in mice attenuates hypoxia-induced pulmonary hypertension [57]. However, hypoxia downregulates TLR4 expression, and TLR4 knockout mice spontaneously develop PH, suggesting that the role of TLR4 is protective in PAH [58]. These results suggest that LPS-TLR4 plays a pivotal role in PAH development (Figure 2). Given the importance of LPS, a pathophysiological link may exist between gut dysbiosis and PAH.

### 2.4. Gut Microbial Metabolites and Pulmonary Arterial Hypertension

Another mechanism by which the microbiota participates in PAH progression is through metabolites. Most nutrients are digested and absorbed in the small intestine, while gut bacteria ferment undigested dietary fibers, proteins, and peptides in the cecum and colon. Microbiota-derived metabolites, including SCFAs, tryptophan (5-HT), and TMAO, have received much attention for their roles in diseases [13,59,60].

### 2.5. Short-Chain Fatty Acids

Short-chain fatty acids are derived from indigestible dietary fibers in the intestines and are the major products of microbial fermentative activity in the gut [61]. The principal SCFAs in the gut include acetate, propionate, and butyrate, whose concentrations are in a ratio of 3:1:1, respectively [62]. Short-chain fatty acids are widely distributed throughout the intestines, with acetate and propionate in the small and large intestines and butyrate mostly in the colon and cecum [63]. Several factors contribute to SCFA production, including diet, GM diversity, and the number of commensal bacteria [63]. Certain gut bacteria regulate the microbial conversion of undigested fibers to SCFAs. Under the action of enteric bacteria, pyruvate produces acetate via the acetyl-CoA pathway, while several Firmicutes species generate butyrate via the acetyl-CoA pathway. Bacteroidetes produce propionate via the succinate pathway, and Firmicutes produce propionate via the lactate pathway [64]. A few gut bacteria, including *Faecalibacterium prausnitzii*, *Eubacterium (E.) rectale*, *E. hallii*, and *Ruminococcus (R.) bromii*, participate in most butyrate production [65]. Resistant starch fermentation, regulated by *R. bromii*, greatly promotes butyrate production in the colon [66]. Colonic epithelial cells absorb most SCFAs, which are transported to the liver via the portal vein and metabolized by hepatocytes. A few SCFAs pass through the liver, and a low but measurable concentration can be detected in the systemic circulation [67]. After entering circulation, SCFAs affect metabolism and the functions of peripheral tissues, such as the lungs. They also regulate immune responses and inflammation, which may affect PAH development [63,68].

Short-chain fatty acids regulate immune responses and inflammation via two pathways. One is the activation of free fatty acid 2 (FFA2), FFA3, and GPR109A receptors, and the other is the inhibition of the generation of histone deacetylases (HDACs) [67].

The human genome encompasses 800 G-protein-coupled receptors (GPRs), and a cluster of four GPR genes (GPR40–GPR43) are closely associated with SCFAs. These GPRs are also called free fatty acid receptors (FFARs) because they can sense free fatty acids. GPR41 and GPR43 have been renamed FFAR2 and FFAR3, respectively [61]. G-protein-coupled receptor 41 and GPR43 are differentially expressed on cell membranes and are involved in diverse cellular functions [69,70]. GPR41 is mainly expressed in blood vessels on ECs, and GPR43 is mainly expressed on immune cells. GPR41 is also expressed on immune cells but at lower levels than GPR43 [71]. SCFAs can regulate immune responses by activating GPRs (e.g., neutrophils and macrophages), which are expressed on most immune cells. For example, in models of colitis, arthritis, and asthma, GPR43 deficiency boosted the synthesis of inflammatory mediators and recruitment of immune cells, thus promoting inflammatory responses [72].

HDACs are a group of deacetylating enzymes that remove acetyl groups from both histone and non-histone protein complexes to regulate gene expression [73]. Butyrate and propionate have key roles in suppressing the production of class I and class IIa HDACs and downregulating silence information regulator 1 expression [74]. By specifically inhibiting HDAC1 and HDAC3, butyrate and propionate are non-competitive HDAC inhibitors [67].

Histone deacetylase inhibitors are potent agents for reducing inflammatory activities in inflammatory diseases [75,76]. For example, HDAC3 promotes the expression of inflammatory genes and recruits monocytes to inflammation sites by regulating NF-kB activity [77]. Suberoylanilide hydroxamic acid, a potent HDAC inhibitor with anti-inflammatory properties, attenuates LPS-induced expression of NF-κB-regulated cytokines [78]. Acetate-producing bacteria are decreased in PAH. One study found notable differences in SCFA-producing bacterial genera between controls and patients with PAH. Acetate levels are also reduced in the serum of PAH-induced rats. This change parallels the differences observed in the types of acetate-producing bacteria [29].

Therefore, SCFAs affect PAH via several mechanisms. Firstly, reduced SCFAs weaken gut barrier functions and favor oxidative metabolism, which further increases the likelihood of gut inflammation and leakiness. In fact, SCFA-producing bacteria, such as *Coprococcus*, *Butyrivibrio*, and *Lachnospiraceae*, are decreased, and gut permeability is increased in patients with PAH [31]. As mentioned, the mucosal layer is the primary physical barrier against pathogen invasion and is mainly composed of mucins secreted by goblet cells in the epithelium. SCFAs are closely associated with mucins. In vitro, SCFAs promoted goblet cells to produce mucins by increasing mucin gene expression [79]. In vivo, administering probiotic supplements boosted SCFA production and markedly enhanced intestinal barrier functions in mice with antibiotic-induced dysbiosis [80]. Short-chain fatty acids also protect against endothelial barrier dysfunction. Tight junctions (TJs) combine with epithelial cells to regulate paracellular transport. Lowered expression of TJ proteins (i.e., occludin, claudin, JAMs, and ZO-1) increases gut permeability. Researchers have demonstrated that SCFAs strengthen barrier functions by regulating TJ protein expression in several disease models [80,81,82,83]. Moreover, SCFAs affect both intestinal and vascular ECs [20]. Short-chain fatty acids upregulated the expression of TJ proteins (ZO-1, occludin, and claudin-1) and a cytoplasmic adaptor protein (cingulin) in rhesus monkey vascular ECs to protect the integrity of the vascular epithelial barrier.

Secondly, SCFAs exert anti-inflammatory effects on chemotaxis and leukocyte recruitment; these effects have been demonstrated in in vitro experiments and animal models [84]. SCFAs activate or induce differentiation of T-regulatory cells either via activation of GPCRs or epigenetic modifications by inhibiting HDACs [85,86]. SCFAs block the NF-kB pathway in macrophages to arrest inflammatory responses and promote T-cell differentiation to enhance IL-10 production [71]. SCFAs boost the generation of colonic T-regulatory cells [87] and attenuate hypoxia-induced pulmonary vascular remodeling by regulating cytokine production in the lungs [20]. In conclusion, SCFAs play unique roles in PAH development; however, the specific molecular mechanisms require further investigation.

### 2.6. Trimethylamine N-Oxide

The GM transforms dietary nutrients, such as phosphatidylcholine, choline, L-carnitine, and betaine from beef and fish, into trimethylamine. Eight bacterial species involved in trimethylamine production have been identified and belong to *Firmicutes* and *Proteobacteria* [88]. Most trimethylamine enters the circulatory system and is subsequently oxidized to TMAO by hepatic flavin-containing monooxygenase, while excess TMA is directly decomposed into dimethylamine or methane [89]. Therefore, the GM composition notably affects TMAO production from dietary sources [90].

Trimethylamine N-oxide is a risk factor for several cardiovascular diseases such as hypertension, heart failure, and atherosclerosis [13,14,91]. Recent research revealed that bacterial communities producing TMAO metabolites were increased in PAH patients and animals [29,31,32]. Because TMAO synthesis relies on the intestinal microbial flora, changes in the GM directly alter circulating TMAO levels.

Trimethylamine N-oxide levels were elevated in high-risk patients with idiopathic PAH and MCT-induced rat models but not in low-risk patients with idiopathic PAH or hypoxia-induced mouse models, indicating that TMAO expression was increased only in severe PAH [21]. Furthermore, 3,3-dimethyl-1-butanol (DMB), a TMAO synthesis inhibitor, alleviated the pathological changes in PAH. DMB treatment reduced right ventricular systolic pressure, maximum right atrial pressure, right ventricular hypertrophy, and pulmonary vascular thickness in MCT-induced rat models [21], suggesting that TMAO plays an important role in accelerating PAH progression and may be a therapeutic target for treating PAH.

Regarding its mechanism, Wang et al. found that DMB prevented macrophages from secreting inflammatory factors and thereby relieved pulmonary vascular damage [21]. It was also noticed that TMAO significantly intensified PASMC proliferation and macrophage migration, which are important contributors to PAH development and progression [21].

### 2.7. 5-Hydroxytryptophan

Serotonin (5-hydroxytryptophan [5-HT]), a natural vasoconstrictor and vascular SMC mitogen, is essential in the PAH pathobiology. Patients with both idiopathic and secondary pulmonary hypertension exhibit increased plasma serotonin levels [92]. Tryptophan is the only substrate for serotonin synthesis. Tryptophan hydroxylase (TPH), the rate-limiting enzyme in serotonin biosynthesis, converts tryptophan to the intermediate, L-5-hydroxytryptophan (5-HTP), which is then transformed into 5-HT. There are two active TPH isoforms: TPH1 and TPH2. A majority of peripheral serotonin is synthesized by TPH1 in the gut, while TPH2 participates in the manufacture of central serotonin and is mostly expressed in the central nervous system. Over 99% of total 5-HT in the circulatory system is taken up by platelets after being released from the intestines. The rate of peripheral 5-HT synthesis is limited by TPH1 levels, which directly affect the plasma serotonin concentration. Previous studies have demonstrated that the GM plays a role in the regulation of 5-HT metabolism in the colon, which in turn might affect peripheral blood concentration of serotonin [93,94,95]. The GM promotes 5-HT biosynthesis by elevating TPH1 expression in colonic ECs and regulate both colon and serum levels of 5-HT.

Serotonin, a natural vasoconstrictor and vascular SMC mitogen, is essential in the PAH pathobiology. Serotonin acts on the pulmonary vasculature via interactions with GPRs (5-HT 1B, 2A and 2B). Substantial evidence suggests that activation of 1B-R, 2A-R, and 2B-R subtypes modulates pulmonary vascular tone. In rodent models, the 5-HT2B receptor has therapeutic effects on heritable PAH owing to its antagonism [96]. In experimental PAH, 5-HT2B receptors play crucial roles in the development of endovascular injury caused by bone marrow-derived proangiogenic cells [97].

The effect of 5-HT1B receptors is achieved via two signaling pathways after activation. First, activated 5-HT receptors induce the expression of downstream RhoA [98]. As a member of the Rho family of small GTPases, RhoA regulates various cellular responses, including contraction, migration, and growth, as well as gene expression and differentiation [99]. RhoA is activated by exchanging GDP for GTP and translocates to the plasma membrane to stimulate downstream effectors such as Rho-kinase. Rho-kinase activation inhibits the expression of myosin light-chain phosphatase by phosphorylating the myosin-binding subunit of myosin light-chain phosphatase. Consequently, Ca^2+^ becomes more sensitive to contraction in vascular SMCs. Furthermore, cytosolic Ca^2+^ increases contraction and myosin light-chain kinase activity [100]. RhoA/Rho-kinase affects nuclear translocation of ERK1/ERK2 during 5HT1BR-mediated mitogenesis [101]. Therefore, GM-serotonin-RhoA/ROCK signaling exerts important roles in the pathogenesis of pulmonary hypertension.

Second, serotonin induces cellular Src-related kinase-regulated Nox1-induced reactive oxygen species (ROS) and nuclear factor (erythroid-derived 2)-like 2 (Nrf-2) dysregulations by activating 5-HT1B receptors. Consequently, post-translational oxidative protein modification is increased, and redox-sensitive signaling pathways are activated. ROS production in the lungs contributes to experimental pulmonary hypertension [102].

The serotonin transporter (SERT or 5-HTT) is a monoamine transporter protein that delivers serotonin into cells. Evidence suggests that 5-HT’s mitogenic effect depends on SERT activity [103]. Transgenic mice with SERT overexpression were more susceptible to pulmonary hypertension [104], and administering SERT inhibitors prevented hypoxia-induced pulmonary vascular remodeling in wild-type animals [105]. Similarly, the 5-HTT inhibitors, fluoxetine and citalopram, impeded human PASMC growth in vitro by blocking SERT expression [106]. miR-361-3p overexpression suppressed serotonin-induced human PASMC proliferation by lowering SERT levels [107], indicating that SERT induced PASMC proliferation. Because SERT mediates indolamine uptake in SMCs, the mitogenic effect of 5-HT on PASMCs is ascribed to 5-HT internalization by SERT. Serotonin entering the cytoplasm of PASMCs is likely decomposed by MAO-A or NADPH oxidases into ROS.

## 3. Gut Microbiota-Directed Therapies for Pulmonary Arterial Hypertension

Considering the important contribution of GM in PAH development described earlier, reconstructing the imbalanced gut by probiotics, prebiotics, or fecal microbiota transplantation may be effective therapeutic strategies for PAH.

### 3.1. Prebiotics and Probiotics

Probiotics are live microorganisms that benefit host health when administered in adequate amounts. Probiotics with non-strain-specific claims include *Bifidobacterium adolescentis, B. animalis, B. bifidum, B. breve, B. longum, Lactobacillus acidophilus, L. casei, L. fermentum, L. gasseri, L. johnsonii, L. paracasei, L. plantarum, L. rhamnosus*, and *L. salivarius* [108]. Wedgwood et al. showed that rat pups with PAH induced by postnatal growth restriction presented gut dysbiosis that was attenuated by administering the probiotic *L. reuteri* DSM 17938 [109]. *L. reuteri* reduces proinflammatory cytokine levels and simultaneously blocks TLR4 signaling via the NF-κB pathway [110]. In children with acute lung injury, a probiotic formulation of *Eosinophil L.* and *L. acidophilus* significantly improved pulmonary artery pressure [111]. Thus, PAH is at least partly driven by dysbiosis in the developing gut, and combining probiotics with established approaches may provide a more effective therapy for treating PAH. However, precise probiotic strains that are both safe and effective in lowering pulmonary arterial pressure and preventing PAH remains a great challenge in clinical application and is worth exploring.

Prebiotics are nondigestible carbohydrates that modulate microbiota activity [112]. Studies show that they selectively enhance the growth of beneficial bacteria such as *Lactobacillus* and *Bifidobacterium*. They also exert a strong anti-inflammatory effect to deter harmful intestinal pathogens. Inulin is a prebiotic that promotes beneficial bacterial growth and stimulates SCFA production. In a randomized controlled trial, dietary inulin or inulin propionate supplementation attenuated propionate release from the colon in obese patients [113]. Similarly, obese patients can lose weight on an inulin-enriched diet [114]. However, no study to date has demonstrated the role of prebiotics in PAH.

### 3.2. Fecal Microbiota Transplantation

Fecal microbiota transplantation (FMT) involves transferring functional bacteria from healthy donor feces into the gastrointestinal tracts of patients to repair an imbalanced GM. Filtered feces collected from healthy donors or recipients (autologous FMT) are transplanted into the intestines of patients with certain diseases. Fecal microbiota transplantation from ACE2-knock-in mice to wild-type mice protected wild-type mice from hypoxia-induced cardiopulmonary pathophysiology and mitigated gut pathology [36]. However, although FMT is currently a reliable treatment for patients with *Clostridioides* difficile colitis [115], the safety (especially in the long term) and efficacy of microbiota-targeting interventions to treat PAH remain uncertain. The U.S. Food and Drug Administration recently approved a phase I trial evaluating the safety and feasibility of FMT in treating PAH [116]. In this open-label trial, 12 patients diagnosed with PAH will take once-daily FMT capsules for 7 days to determine whether FMT is safe and feasible for these patients. This is the first step in investigating the effectiveness of FMT as a treatment for PAH. As FMT is more effective than probiotics or prebiotics in reconstructing the GM, the therapeutic effect of FMT for PAH may be better; however, large-scale, multicenter clinical studies are needed to obtain a definitive conclusion.

## 4. Conclusions

Gut dysbiosis and PAH form a circularity interaction. A shift in gut microbiota composition causes an increase in gut permeability, thereby contributing to bacteria translocation and metabolic endotoxemia. Gut dysbiosis also changes the circulatory levels of gut metabolites such as TMAO, 5-HT, and SCFAs. Collectively, endotoxin and metabolic products exert on pulmonary vasculature, causing pulmonary perivascular inflammation, pulmonary vascular remodeling, and eventually developing the PAH. Meanwhile, long-term PAH leads to intestinal vein congestion, which in turn contributes to gut dysbiosis. However, more research is needed to investigate and verify an etiological link between the PAH pathogenesis and changes in the gut microbiota. Reconstructing the GM by prebiotics, probiotics, or FMT may provide new effective therapeutic strategies for PAH. Nevertheless, it is imperative to uncover the specific bacterial species and metabolic pathways in the context of PAH in future studies. 

## Figures and Tables

**Figure 1 nutrients-14-04278-f001:**
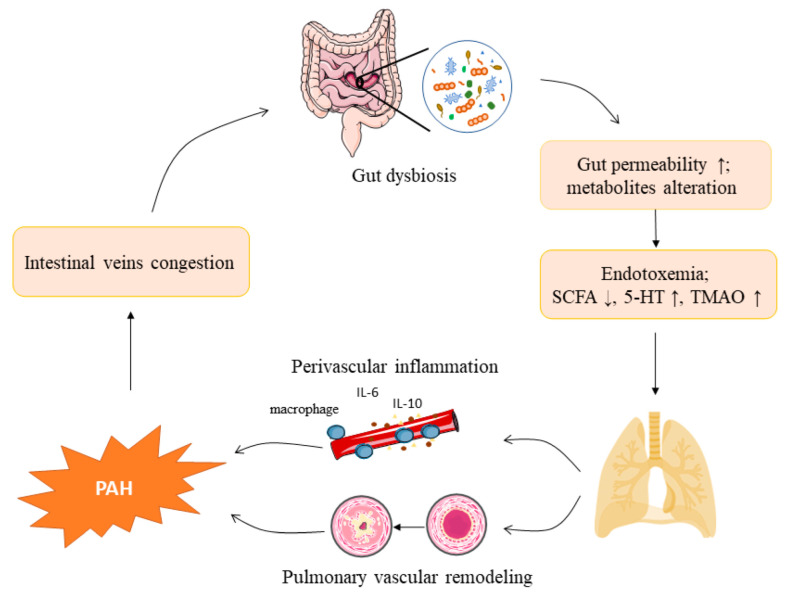
The roles and underlying mechanisms of gut microbiota in pulmonary arterial hypertension. A shift in gut microbiota composition causes an increase in gut permeability, which leads to bacteria translocation and metabolic endotoxemia. Gut dysbiosis also changes the circulatory levels of gut metabolites such as TMAO, 5-HT, and SCFAs. Collectively, endotoxin and metabolic products exert on pulmonary vasculature, causing pulmonary perivascular inflammation, pulmonary vascular re-modeling, and eventually PAH. Long-term PAH leads to intestinal vein congestion, which in turn contributes to gut dysbiosis. TMAO: trimethylamine N-oxide; 5-HT: 5-hydroxytryptophan; SCFAs: short-chain fatty acids; IL-6: interleukin-6; IL-10: interleukin-10; PAH: pulmonary arterial hypertension, ↑ increased; ↓ decreased.

**Figure 2 nutrients-14-04278-f002:**
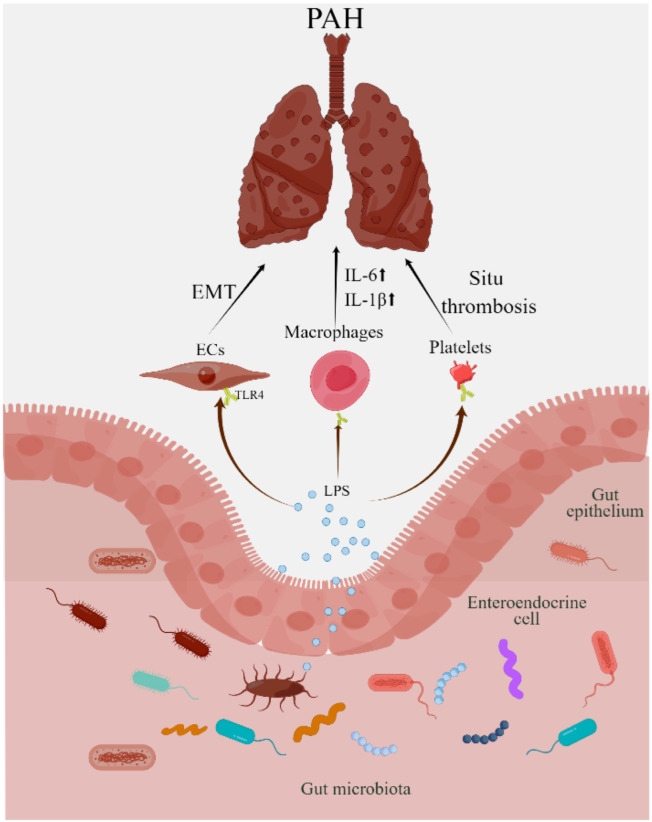
Increased gut permeability leads to translocation of gut bacterial products, such as LPS, into circulation. LPS is recognized by TLR4 expression in endothelial cells, macrophages, and platelets. After blinding to TLR4, LPS promotes endothelial–mesenchymal transition, macrophage inflammatory reaction, platelet in situ thrombosis, and finally PAH. LPS: lipopolysaccharide; TLR4: Toll-like receptor 4, ↑ increased.

**Table 1 nutrients-14-04278-t001:** Shifts in the gut microbiota composition in pulmonary arterial hypertension.

Experimental Models/Populations	Effects and Observation	Preclinical or Clinical	Ref.
SUGEN5416/hypoxia rat model	↑Firmicutes, Actinobacteria, Cyanobacteria ↓Bacteroides, Akkermansia	**Preclinical**	Takayuki J. Sanada et al. (2020) [35]
MCT-induced rat model	↑Firmicutes, Proteobacteria, Actinobacteria ↑Allobaculum, Ralstonia, Bifidobacterium ↓Bacteroidota, Spirochaetota ↓Lactobacillus, Romboutsia	**Preclinical**	Wei Hong et al. (2021) [30]
mice with hypoxia	↑F/B, Proteobacteria, Prevotella, Oscillospira, Ruminococcus ↓Lactobacillus	**Preclinical**	Ravindra K. Sharma et al. (2020) [36]
SUGEN5416/hypoxia rat model	↓Bacteroidetes, Odoribacteraceae ↑Firmicutes, Peptostreptococcaceae	**Preclinical**	María Callejo et al. (2018) [29]
Patients with type I PAH	↑S. parasanguinis, R. gnavus, C. aerofaciens ↑Collinsella, Blautia ↓B. crossotus, B. cellulosilyticus, E. siraeum, B. vulgatus, A. muciniphila, ↓B. intestinihominis	**Clinical**	Seungbum Kim et al. (2020) [31]
Patients with CTEPH	↓Faecalibacterium, Roseburia, Fusicatenibacter	**Clinical**	Yumiko Ikubo et al. (2022) [32]

Abbreviations: MCT, monocrotaline; PAH, pulmonary arterial hypertension; CTEPH, chronic thromboembolic pulmonary hypertension; F/B, Firmicutes/Bacteroides, ↑ increased; ↓ decreased.

## Data Availability

Not applicable.

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
