# Peer review of "The Role and Mechanism of Gut Microbiota in Pulmonary Arterial Hypertension"

_nutrients, 2022, doi:10.3390/nu14204278_

Round 1
Reviewer 1 Report
The authors reviewed the effect of gut microbiota on pulmonary arterial hypertension. The manuscript content falls into the special issue aims. Although the considered title is interesting, there are some concerns and comments which should be carefully considered to prepare a better version during the revision.
1. The first issue is the language level which is not appropriate for this review. There are a lot of grammatical and punctual errors. The authors must revise the text for better readability.
2. The combined name ‘’gut microbiota’’ was used about 49 times in the manuscript. It could have been abbreviated as ‘GM’ in the abstract and text.
3. More Keywords should be added. Add the name of mechanisms and avoid bringing metabolites and intermediates affecting in the control of pulmonary arterial hypertension.
4. The introduction is truly short and should be improved by adding more references. A strong literature review in this section can show the novelty of the present review. As a reviewer, I am not satisfied with this introduction because I did not find any answer to the question why did you review this subject? Thus, give some literature in the introduction, and present the study gaps to prove the necessity and importance of writing this overview.
5. Do not use ‘’the abbreviated forms’’ in subtitles such as ‘’ 2.3 LPS in PAH’’. Check carefully and correct.
6. The manuscript has only two figures (writing Figure.2 is wrong in the text). It seems that the authors brought them just for having some illustrations in the manuscript because there was no deep discussion of the introduced mechanisms. Also, the use of at least one or two tables could shorten the text of this review by including different functions.
7. The authors must abbreviate the name of microorganisms after the first time, for example, on page 7, Eubacterium rectale, E. hallii, and Ruminococcus bromii,…and then at the end of the second line, it should be R. bromii…, this problem should be carefully check in throughout the text.
8. In general, the manuscript has a high number of abbreviated forms, which makes it difficult to follow some results. Also, the authors should not use abbreviated forms in the initial of the sentence.
9. The part of 3.3 5-hydroxytryptophan does not have any content integrity while it shows some separated articles without any content association.
10. The section of ‘’Gut Microbiota-Directed Therapies for PAH’’ just is a summary of limited related studies, while there is no significant discussion on the function of gut microbiota such as prebiotics and probiotics, etc.
11. The section on conclusions should be rewritten. It was badly written. Some concerns in this part include i) it never covers the overall results of the present review, ii) it should not be in two paragraphs unless the future directions/recommendations will be added (which should be in this part), iii) the use of the term ‘’ Several studies have confirmed that …’’ in this section is wrong. You should have reviewed all studies before and thus conclusive sentences should be included in this section, iv) do not use abbreviated forms in the initial of sentences like PAH at the beginning of the conclusions, etc.
12. References at all are not by the guidelines of the MDPI journal. All of them should be justified.
Author Response
The authors reviewed the effect of gut microbiota on pulmonary arterial hypertension. The manuscript content falls into the special issue aims. Although the considered title is interesting, there are some concerns and comments which should be carefully considered to prepare a better version during the revision.
1. The first issue is the language level which is not appropriate for this review. There are a lot of grammatical and punctual errors. The authors must revise the text for better readability.
The author’s answer: Thanks for the reviewer's professional comment. Efforts were made to correct the spelling and grammar mistakes in the revised manuscript, and changes were all highlighted within the document.
2. The combined name ‘’gut microbiota’’ was used about 49 times in the manuscript. It could have been abbreviated as ‘GM’ in the abstract and text.
The author’s answer: Thanks so much for pointing out the redundant words. As suggested, ‘gut microbiota’ has been replaced by ‘GM’ in the revised manuscript.
3. More Keywords should be added. Add the name of mechanisms and avoid bringing metabolites and intermediates affecting in the control of pulmonary arterial hypertension.
The author’s answer: Thanks for the reviewer's valuable suggestion. We have changed keywords ‘gut microbiota; short-chain fatty acids; lipopolysaccharide’ to ‘gut microbiota dysbiosis; pulmonary vascular remodeling; metabolism; prebiotics and probiotics; microbiota transfer therapy’ in order to get closer to the subject.
4. The introduction is truly short and should be improved by adding more references. A strong literature review in this section can show the novelty of the present review. As a reviewer, I am not satisfied with this introduction because I did not find any answer to the question why did you review this subject? Thus, give some literature in the introduction, and present the study gaps to prove the necessity and importance of writing this overview.
The author’s answer: Thanks for the reviewer's professional comment. The introduction section was added with more literature to show the novelty and importance of this review.
5. Do not use ‘’the abbreviated forms’’ in subtitles such as ‘’ 2.3 LPS in PAH’’. Check carefully and correct.
The author’s answer: Thanks for the reviewer's kindly reminding. Abbreviations throughout the manuscript were checked carefully and corrected correspondingly.
6. The manuscript has only two figures (writing Figure.2 is wrong in the text). It seems that the authors brought them just for having some illustrations in the manuscript because there was no deep discussion of the introduced mechanisms. Also, the use of at least one or two tables could shorten the text of this review by including different functions.
The author’s answer: Thanks for the reviewer's valuable suggestion. Figure1 was redrawn to demonstrate the circularity of gut dysbiosis and PAH, and one table was added to shorten the text of the review.
7. The authors must abbreviate the name of microorganisms after the first time, for example, on page 7, Eubacterium rectale, E. hallii, and Ruminococcus bromii, and then at the end of the second line, it should be R. bromii…, this problem should be carefully check in throughout the text.
The author’s answer: Thanks for the reviewer's valuable suggestion. The name of microorganisms was checked carefully throughout the text, and was abbreviated after the first time.
8. In general, the manuscript has a high number of abbreviated forms, which makes it difficult to follow some results. Also, the authors should not use abbreviated forms in the initial of the sentence.
The author’s answer: Thanks for the reviewer's comment. Unnecessary abbreviations were deleted, and abbreviated forms in the initial of the sentences were corrected in the revised manuscript.
9. The part of 3.3 5-hydroxytryptophan does not have any content integrity while it shows some separated articles without any content association.
The author’s answer: Thanks for the reviewer's comment. The part of 3.3 5-hydroxytryptophan was revised to show the content integrity and association.
10. The section of ‘’Gut Microbiota-Directed Therapies for PAH’’ just is a summary of limited related studies, while there is no significant discussion on the function of gut microbiota such as prebiotics and probiotics, etc.
The author’s answer: Thanks for the reviewer's comment. Discussion on the function of gut microbiota such as prebiotics, probiotics, and fecal microbiota transplantation was added in the revised manuscript.
11. The section on conclusions should be rewritten. It was badly written. Some
concerns in this part include i) it never covers the overall results of the present review, ii) it should not be in two paragraphs unless the future directions/recommendations will be added (which should be in this part), iii) the use of the term ‘’ Several studies have confirmed that …’’ in this section is wrong. You should have reviewed all studies before and thus conclusive sentences should be included in this section, iv) do not use abbreviated forms in the initial of sentences like PAH at the beginning of the conclusions, etc.
The author’s answer: Thanks for the reviewer's professional comments and suggestions. The section of conclusions was rewritten and concerns of the reviewer were avoided in the revised manuscript.
12. References at all are not by the guidelines of the MDPI journal. All of them should be justified.
The author’s answer: Thanks for the reviewer's comment. The references were revised according to the guidelines of the MDPI journal.

Reviewer 2 Report
This aim of this mini-review is to provide an overview of the potential role of the gut microbiome as both a causative agent in the development of pulmonary arterial hypertension (PAH), as well as a driving force behind the pathological manifestations of PAH. The manuscript is generally well-written but could be organized a little better. Specifically, it is very “choppy” and fails to string the information into a flowing narrative. At times, it is also unclear what point is being made, especially the links between individual metabolites and pulmonary function. This is partly due to effects on other cell types being intertwined into the story without directly tying it back into pulmonary function.
Given the time devoted to discussing the potential role of LPS, and TLR-4 signaling, I am surprised there is no discussion of TLR-4 deficient models and PAH since these data are available.
Figure 1 should be redrawn to demonstrate the circularity of gut dysbiosis and PAH more clearly. That is, that PAH can lead to gut dysbiosis which can then further aggravate PAH. This is gleaned from the text easily enough, but it is not apparent from looking at Figure 1. This would also eliminate the need for Figure 2.
Given that the ms opens with the societal impact of PAH and a call for more research into curative therapies, I would have expected more discussion regarding pre- and probiotic interventions in addition to fecal transplant therapies as a curative therapy for PAH. The material provided is very superficial even if studies that have looked at the role of each of these approaches in PAH specifically are few. Likewise for fecal transplantation.
Round 2
Reviewer 1 Report
There are still some comments to improve the content as follows:
1. Table 1: it is incorrect that the authors assigned two columns for researchers' names (1st column) and ref. (the end column). Regarding the format of MDPI journals, the first column should be removed. It is also recommended that the authors add a new column for a new factor/variable which is different among the selected studies to have a more comprehensive table.
2. Inclusion of Figure 2 in the conclusions section is frustrating. The authors should find another vacancy to include this figure and discuss/elaborate on this figure for other researchers. Apart from this point, the use of this figure in the conclusions section actually provided a barrier to providing more information.
3. In references, the cited article title should be written with an uppercase letter in the initial for all words (nouns, pronouns, and adjectives). Check carefully all of the cited articles on their title.
4. Please recheck the language carefully because some mistakes can be easily found in the text.
Author Response
- Table 1: it is incorrect that the authors assigned two columns for researchers' names (1st column) and ref. (the end column). Regarding the format of MDPI journals, the first column should be removed. It is also recommended that the authors add a new column for a new factor/variable which is different among the selected studies to have a more comprehensive table.
The author’s answer: Thanks for the reviewer's valuable suggestion. We have added the 1st column (researchers' names) the end column to (ref.) and added a new column to make a more comprehensive table.
- Inclusion of Figure 2 in the conclusions section is frustrating. The authors should find another vacancy to include this figure and discuss/elaborate on this figure for other researchers. Apart from this point, the use of this figure in the conclusions section actually provided a barrier to providing more information.
The author’s answer: Thanks for the reviewer's comment. We have shifted Figure 2 to the introductions section to increase the clarity of the figure.
- In references, the cited article title should be written with an uppercase letter in the initial for all words (nouns, pronouns, and adjectives). Check carefully all of the cited articles on their title.
The author’s answer: Thanks so much for pointing out the cited article title without capital letter. As suggested, we have uniformed the titles as first letter capitalized in the whole references.
- Please recheck the language carefully because some mistakes can be easily found in the text.
The author’s answer: Thanks for the reviewer's professional comment. Efforts were made to correct the spelling and grammar mistakes in the revised manuscript, and changes were all highlighted within the document.
Reviewer 2 Report
The authors have adequately addressed each of my comments. No further revision is needed.
Author Response
The author’s answer: Thanks for the reviewer's professional comment.